# Primary and hTERT-Transduced Mesothelioma-Associated Fibroblasts but Not Primary or hTERT-Transduced Mesothelial Cells Stimulate Growth of Human Mesothelioma Cells

**DOI:** 10.3390/cells12152006

**Published:** 2023-08-05

**Authors:** Alexander Ries, Astrid Slany, Christine Pirker, Johanna C. Mader, Doris Mejri, Thomas Mohr, Karin Schelch, Daniela Flehberger, Nadine Maach, Muhammad Hashim, Mir Alireza Hoda, Balazs Dome, Georg Krupitza, Walter Berger, Christopher Gerner, Klaus Holzmann, Michael Grusch

**Affiliations:** 1Center for Cancer Research and Comprehensive Cancer Center, Medical University of Vienna, Borschkegasse 8a, 1090 Vienna, Austria; alexander.ries@meduniwien.ac.at (A.R.); christine.pirker@meduniwien.ac.at (C.P.); doris.mejri@meduniwien.ac.at (D.M.); thomas.mohr@mohrkeg.co.at (T.M.); karin.schelch@meduniwien.ac.at (K.S.); dflehberger@gmail.com (D.F.); a51840311@unet.univie.ac.at (N.M.); mhashim@bs.qau.edu.pk (M.H.); walter.berger@meduniwien.ac.at (W.B.); klaus.holzmann@meduniwien.ac.at (K.H.); 2Department of Analytical Chemistry, University of Vienna, Waehringer Straße 38, 1090 Vienna, Austria; astrid.slany@univie.ac.at (A.S.); johanna.c.mader@univie.ac.at (J.C.M.); christopher.gerner@univie.ac.at (C.G.); 3Joint Metabolome Facility, University of Vienna and Medical University of Vienna, Waehringer Guertel 38, 1090 Vienna, Austria; 4ScienceConsult—DI Thomas Mohr KG, Enzianweg 10a, 2353 Guntramsdorf, Austria; 5Department of Thoracic Surgery, Medical University of Vienna, Waehringer Guertel 18-20, 1090 Vienna, Austria; mir.hoda@meduniwien.ac.at (M.A.H.); balazs.dome@meduniwien.ac.at (B.D.); 6National Korányi Institute of Pulmonology, Korányi Frigyes u. 1, 1122 Budapest, Hungary; 7Department of Thoracic Surgery, National Institute of Oncology, Semmelweis University, Rath Gyorgy u. 7-9, 1122 Budapest, Hungary; 8Department of Translational Medicine, Lund University, Sölvegatan 19, 22184 Lund, Sweden; 9Department of Pathology, Medical University of Vienna, Waehringer Guertel 18-20, 1090 Vienna, Austria; georg.krupitza@meduniwien.ac.at

**Keywords:** pleural mesothelioma, pleural mesothelial cells, mesothelioma-associated fibroblasts, tumor microenvironment, conditioned medium, human telomerase reverse transcriptase, hTERT

## Abstract

Pleural mesothelioma (PM) is an aggressive malignancy that develops in a unique tumor microenvironment (TME). However, cell models for studying the TME in PM are still limited. Here, we have generated and characterized novel human telomerase reverse transcriptase (hTERT)-transduced mesothelial cell and mesothelioma-associated fibroblast (Meso-CAF) models and investigated their impact on PM cell growth. Pleural mesothelial cells and Meso-CAFs were isolated from tissue of pneumothorax and PM patients, respectively. Stable expression of hTERT was induced by retroviral transduction. Primary and hTERT-transduced cells were compared with respect to doubling times, hTERT expression and activity levels, telomere lengths, proteomes, and the impact of conditioned media (CM) on PM cell growth. All transduced derivatives exhibited elevated hTERT expression and activity, and increased mean telomere lengths. Cell morphology remained unchanged, and the proteomes were similar to the corresponding primary cells. Of note, the CM of primary and hTERT-transduced Meso-CAFs stimulated PM cell growth to the same extent, while CM derived from mesothelial cells had no stimulating effect, irrespective of hTERT expression. In conclusion, all new hTERT-transduced cell models closely resemble their primary counterparts and, hence, represent valuable tools to investigate cellular interactions within the TME of PM.

## 1. Introduction

Pleural mesothelioma (PM) is a highly lethal malignancy arising from the neoplastic growth of mesothelial cells that form the lining layers of the pleural cavity [1]. Late diagnosis and limited treatment options contribute to a very poor median survival rate of little more than 12 months [2]. PM is genetically characterized by recurrent inactivating mutations or deletions of tumor suppressor genes, such as BRCA1-associated protein 1 (BAP1), CDKN2A, or TP53 [1]. Driver mutations in oncogenes, like Ras or epidermal growth factor receptor (EGFR), in contrast, are uncommon in PM. Rapid local tumor spreading and frequent recurrence after treatment, in the absence of oncogenic aberrations, indicate a support of tumor development and progression by stimulating signals from cells of the tumor microenvironment (TME). Etiologically, the development of PM is mainly associated with asbestos exposure [3] and characterized by a long latency period of up to 50 years [4]. However, it has also been associated with exposure to various other types of naturally occurring mineral fibers that exhibit asbestos-like characteristics, such as erionite or fluoro-edenite [5]. Macrophages fail to eliminate the inhaled fibers, accumulate, and create a unique inflammatory environment, leading to the malignant transformation of pleural mesothelial cells [6]. Different cell types of the TME can acquire an activated phenotype in response to cancer-associated inflammatory signals and promote several pro-tumorigenic features including angiogenesis, proliferation, invasion, and metastasis [7]. Activated fibroblasts, called cancer-associated fibroblasts (CAFs), represent one of the main components in the TME of various cancer types and have been shown to regulate many aspects of tumor progression including proliferation, migration, invasion, and therapy response [8]. Primary CAFs isolated from tumor specimens of PM patients, termed mesothelioma-associated fibroblasts (Meso-CAFs), have recently been demonstrated to substantially enhance growth and migration of PM cells via secreted signaling molecules [9]. Continuous exposure to paracrine signaling from activated cells in the vicinity may also contribute to the malignant transformation of mesothelial cells. Indeed, extracellular vesicles released by Meso-CAFs have recently been shown to markedly activate pro-oncogenic signaling pathways in normal untransformed mesothelial cells and promote a pre-neoplastic phenotype [10]. Understanding the complex underlying mechanisms that govern tumor-promoting interactions between cells of the TME, such as pleural mesothelial cells, Meso-CAFs, and tumor cells, may help to identify new molecular targets for emerging preventive or therapeutic strategies in PM. To date, the number of available cell lines for TME research in PM is still very limited and, hence, only a few studies have focused on CAFs of PM, and the generation of an immortalized Meso-CAF cell model has not been described so far. Primary cells of humans generally enter an irreversible state of growth arrest after a limited number of cell divisions. A fundamental mechanism underlying this state of replicative senescence is the progressive shortening of telomeres during each division, which results from a lack of telomerase activity [11]. Immortalized cell models with an extended life span are, therefore, very helpful to study cellular interactions within the TME. Although normal mesothelial cell lines are essentially used as controls in many experiments, the small number of currently available cell lines hampers research progress [12]. The simian virus 40 (SV40)-immortalized cell line Met5A has been the most commonly used pleural mesothelial cell model for over 30 years [13]. Since SV40-mediated immortalization can cause cellular changes, including karyotypic instability, abnormal differentiation, and deficiencies in DNA damage response [14], there is a great need for additional cell models that closely resemble normal pleural mesothelial cells. As an advantage, human telomerase reverse transcriptase (hTERT)-immortalized cells exhibit fewer karyotypic changes and retain more properties of their primary counterpart [14].

Here, we describe hTERT-transduced Meso-CAFs and normal pleural mesothelial cells and investigate their ability to stimulate PM cell growth. The primary cultures of both cell types exhibited clearly distinct gene and protein expression profiles. We demonstrate that the transduction with hTERT did not elicit marked changes of the cells regarding their morphology, proteomes, and impact on PM cell growth. While the conditioned media (CM) of primary and hTERT-transduced Meso-CAFs stimulated PM cell growth to the same extent, the CM of pleural mesothelial cells did not cause a growth stimulating effect irrespective of hTERT transduction.

## 2. Materials and Methods

### 2.1. Isolation of Primary Cells from Human Tissue

The tissue sample used for the isolation of the non-malignant human pleural mesothelial cell culture NP2 was collected from a surgery specimen of a pneumothorax patient following previously described protocols [15,16]. Meso-CAFs from tumor samples of patients diagnosed with PM were isolated as recently described [9]. In both cases, the surgery was performed at the Department of Thoracic Surgery of the Medical University of Vienna, and informed consent to use the material for research purposes was obtained. The study was approved by the ethics committee of the Medical University of Vienna (EK Nr. 904/2009). In brief, the tissue was initially minced and incubated in cell culture flasks (T25) under normoxic cell culture conditions (37 °C, 5% CO_2_). The NP2 and Meso-CAF tissue samples were cultured in ACL (prepared in our group with ingredients listed in Appendix A) and RPMI-1640 growth medium (#R6504, Sigma-Aldrich, St. Louis, MO, USA), respectively, both supplemented with 10% heat-inactivated fetal bovine serum (FBS) (Biowest, France) and antibiotics (100 U/mL penicillin and 100 μg/mL of streptomycin (1% P/S)) (Sigma-Aldrich, St. Louis, MO, USA). Incubation was carried out for one week without agitation of the flasks to allow attachment of tissue pieces to the culture device and outgrowth of cells. The cells were further cultivated, expanded, and regularly washed with sterile phosphate-buffered saline (PBS, 1×) to remove cell debris.

### 2.2. Cell Culture

The primary pleural mesothelial cells NP2 and its hTERT-transduced derivative NP2-hT^+^ were cultured in ACL growth medium supplemented with 10% FBS. The primary Meso-CAFs (Meso109F, Meso125F) and their hTERT-transduced derivatives (Meso109F-hT^+^, Meso125F-hT^+^) as well as the human mesothelioma cell lines (MSTO-211H, SPC212) were all kept in RPMI-1640 medium with 10% FBS. MSTO-211H was purchased from the American Type Culture Collection (ATCC, Rockville, MD, USA) and SPC212 was kindly provided by Prof. R. Stahel (University of Zurich, Zurich, Switzerland). The SV40-immortalized, non-malignant human pleural mesothelial cell line Met5A and both normal primary human lung fibroblasts, MRC-5 and Wi38, were also obtained from ATCC (Rockville, MD, USA) and cultivated in the respective growth medium according to the supplier’s protocol (Met5A in RPMI-1640, MRC-5 in Dulbecco’s Modified Eagle’s Medium (DMEM) (#D5648, Sigma-Aldrich, St. Louis, MO, USA), and Wi38 in Eagle’s minimal essential medium (MEM) (#M5650, Sigma-Aldrich, St. Louis, MO, USA), all supplemented with 10% FBS). The primary pleural mesothelial cells and the primary Meso-CAFs used for experiments were at fewer than 15 passages. All used cells were maintained in a humidified atmosphere (37 °C, 5% CO_2_), regularly passaged by trypsinization, and routinely checked for *Mycoplasma* contamination.

### 2.3. Induction of Stable hTERT-EGFP Expression in Primary Pleural Mesothelial Cells and Primary Mesothelioma-Associated Fibroblasts

Retroviral transduction was performed to introduce permanently elevated hTERT and stable enhanced green fluorescent protein (EGFP) expression in the primary pleural mesothelial cells NP2 and the primary Meso-CAFs Meso109F and Meso125F. A retroviral vector containing the hTERT gene was modified by replacing the puromycin resistance gene with the cDNA of EGFP to enable immediate identification of stably transfected clonal populations by their fluorescence signal. The original retroviral expression vector (pBABE-puro-hTERT) was kindly provided by Professor W.C. Hahn and has been previously published [17]. The generation of retrovirus particles in HEK293 cells and the transduction of target cells was carried out as described in previous publications [9,18]. The HEK293 cells were obtained from ATCC and cultured in DMEM with 10% FBS under regular culture conditions (37 °C, 5% CO_2_). Calcium phosphate co-precipitation-mediated transfection was used to introduce the hTERT-EGFP vector as well as the two helper plasmids pVSV-G (Clontech, Mountain View, CA, USA) and p-gag-pol-gpt [19] into the host cells. The retroviral particles were produced in the transfected HEK293 cells and released into the growth medium supernatant, which was subsequently used for the transduction of primary NP2, Meso109F, and Meso125F. Further passaging resulted in outgrowth of cultures containing hTERT-transduced cells. While untransduced cells slowed down proliferation and became senescent, stable hTERT- and EGFP-expressing cells continuously divided until >90% of the cells in each culture were EGFP-positive.

### 2.4. Cell Authentication by Short Tandem Repeat (STR)

All cell lines were authenticated using short tandem repeat (STR) DNA profiling. DNA was extracted from cell pellets using QIAamp DNA Blood Mini Kit (Qiagen Sciences, Germantown, MD, USA) following the manufacturer’s instructions (QIAamp 96 DNA Blood Handbook, Fourth Edition, April 2010). STR analysis was performed by the cell line typing service of Microsynth Austria GmbH (Vienna, Austria).

### 2.5. Determination of Doubling Times

The doubling times of the cells were evaluated as recently described [9]. The cells were seeded in a 12-well plate and cell numbers were counted in duplicates at different time points using a Neubauer cell counting chamber. Final doubling times were determined from at least four time points of at least three biological replicates and calculated using the following formula: doubling time [h] per time point = (time [h]) × log (2)/(log (cell count at time point) − log (seeded cell number)).

### 2.6. Analysis of Whole-Genome Gene Expression

Sample preparation, expression analysis, and data evaluation were carried out as described in previous publications [9,20]. Total RNA was isolated from cell cultures with a confluence of about 80% using RNeasy Mini Kit (Qiagen Sciences, Germantown, MD, USA) according to the respective protocol (RNeasy Mini Handbook, HB-0435-006, October 2019). Analysis was performed on 4 × 44 K whole genome oligonucleotide-based gene expression microarrays (Agilent, Santa Clara, CA, USA) and a G2600D Micro Array Scanner (Agilent, Santa Clara, CA, USA). Feature extraction was carried out using the Feature Extraction software (version 11.5.1.1, Agilent, Santa Clara, CA, USA). Data of all oligos with gene annotation are shown in Appendix A. Processing of gene expression data and generation of heatmaps was conducted in R using the packages “genefilter” [21] and “ComplexHeatmap” [22,23]. Genes with several oligonucleotide probes on the array were summarized to the probe with the maximal interquartile range.

### 2.7. Determination of hTERT Gene Expression

The expression of hTERT was evaluated using quantitative real-time reverse transcription PCR (qPCR). Total RNA was isolated from the cells using the innuPREP RNA Mini Kit (Analytik Jena, Jena, Germany) and reverse transcribed with the RevertAid RT Kit (Thermo Scientific, Thermo Fisher Scientific, Carlsbad, CA, USA), both according to the respective manufacturer’s instructions (innuPREP RNA Mini Kit, HB_KS-2080_e_170719, July 2017; RevertAid RT Kit User Guide). The qPCR was performed on a C1000 Touch Thermal Cycler (Bio-Rad Laboratories, Hercules, CA, USA) using SYBR Green PCR Master Mix (Applied Biosystems, Thermo Fisher Scientific, Carlsbad CA, USA). Reaction primers with the following sequence were used to detect an amplicon of 149 base pairs: 5′-ACCGGAAGAGTGTCTGGAGCAA-3′, 5′-GGGGATGAAGCGGAGTCTGGA-3′. Glyceraldehyde-3-phosphate dehydrogenase (GAPDH) (5′-AGCTCACTGGCATGGCCTTC-3′, 5′-ACGCCTGCTTCACCACCTTC-3′) and β-actin (5′-ACTCTTCCAGCCTTCCTTC-3′, 5′-GATGTCCACGTCACACTTC-3′) were both used as reference genes for normalization, and the hTERT expression levels were quantified with the ΔCt method.

### 2.8. Determination of Telomerase Activity

The telomerase activity was determined using qPCR telomeric repeat amplification protocol (qPCR-TRAP) with minor modifications and quantified as total product generated (TPG) units, and polyacrylamide gel electrophoresis (PAGE)-TRAP, both described in previous publications [24,25,26,27]. In brief, the proteins were isolated from the cells using CHAPS buffer, and concentrations were determined via Bradford assay (AppliChem, Darmstadt, Germany). The qPCR-TRAP assay was performed on an ABI PRISM 7500 Fast Sequence Detection System (Applied Biosystems, Foster City, CA, USA) using the GoTaq PCR Master Mix (Promega Corporation, Fitchburg, WI, USA), and the following conditions were applied: Incubation for 30 min at 30 °C and 42 subsequent cycles of 30 s at 95 °C and 30 s at 60 °C. Each reaction contained 0.6 μg of protein extract as well as 200 nM of telomerase substrate (TS) and anchored return CX (ACX) primers. The oligonucleotide primers were used as published previously [27]. The relative telomerase activity values were converted into TPG units using a standard curve, which was generated by serial dilutions of the telomerase substrate oligonucleotide TSR8 (with eight telomeric repeats). One TPG unit corresponds to 0.001 amoles (~600 molecules) of TSR8 extended by at least four telomeric repeats within 30 min at 30 °C. The obtained results were additionally validated by PAGE-TRAP using standard PCR [27]. Extracts were considered positive if a typical six base pair DNA ladder could be observed on the gel.

### 2.9. Determination of Absolute Telomere Lengths

The absolute telomere lengths (aTL) were measured by qPCR following the published protocols [28]. In short, the remaining cell debris pellet after protein extraction with CHAPS buffer was used for DNA isolation with the Gentra Puregene Cell Kit (Qiagen Sciences, Germantown, MD, USA). The DNA was quantified with iQuant dsDNA BR Assay Kit (ABP Biosciences, Beltsville, MD, USA) and 10 ng DNA were subjected to qPCR for the measurement of telomere lengths. The reaction was carried out on a C1000 Touch Thermal Cycler (Bio-Rad Laboratories, Hercules, CA, USA) using GoTaq PCR Master Mix (Promega Corporation, Fitchburg, WI, USA) with the respective primers. Two standard curves were generated by including serially diluted standard oligomers and used to calculate the aTL. The telomere standard curve was established using an oligomer with a known number of TTAGGG repeats (14 times) to estimate the content of telomere sequence repeats per sample reaction tube. The gene copy standard curve was generated using the single copy gene 36B4 as control for amplification to correct for changes in template amounts and determine genome copies per sample. All used oligonucleotide primers are described in the protocols [28], and the aTL was calculated in kb telomere per genome.

### 2.10. Analysis of Protein Fractions

The proteins of different cellular fractions (supernatant, cytoplasmic, and nuclear proteins) were analyzed in three biological replicates using mass spectrometry-based proteomics. The protein fractionation and analysis were performed as described in detail [9,29]. In short, the cells were cultured in T25 flasks until exhibiting 80–90% confluence, and the growth medium was exchanged to FBS-free medium 6 h before protein extraction. The supernatants containing the secreted proteins were initially collected and the fractions of cytoplasmic and nuclear proteins were separated by lysing the cells in isotonic lysis buffer, applying mechanical shear stress with a syringe and several centrifugation steps. All protein fractions were individually precipitated in ethanol at −20 °C overnight. The isolated proteins were dissolved in sample buffer (7.5 M urea, 1.5 M thiourea, 4% CHAPS, 0.05% SDS, 100 mM dithiothreitol), and the concentrations were measured using Bradford assay (Bio-Rad Laboratories, Hercules, CA, USA). Before subjecting the samples to LC-MS/MS analyses, the proteins were enzymatically digested to peptides with Trypsin/Lys-C (Promega Corporation, Fitchburg, WI, USA) and cleaned-up on SDB-RPS StageTips. Analysis of peptide samples was conducted in technical replicates on a Dionex Ultimate 3000 nano LC-system (Thermo Scientific, Thermo Fisher Scientific, Carlsbad, CA, USA) coupled to a timsTOF Pro mass spectrometer (Bruker Daltonics, Bruker Corporation, Billerica, MA, USA). Max-Quant (version 1.6.17.0) [30] with the Andromeda search engine was used to identify the proteins in the samples by searching them against the UniProt Database for human proteins (version 12/2019 with 20,380 entries). The mass spectrometry proteomics data were deposited at the ProteomeXchange Consortium via the Proteomics Identification Database (PRIDE) partner repository and can be accessed with the identifiers PXD035987, PXD036017, PXD036127, PXD043511, and PXD043534 [31]. Data processing and visualization were conducted in R with the packages “DEP” [32], “ComplexHeatmap” [22], and “ggplot2” [33], as well as by the use of Venny 2.1 [34] and the software Perseus (version 1.6.14.0) [35,36]. Moreover, gene ontology (GO) enrichment analysis was performed with the DAVID functional analysis tool [37,38] using the KEGG (Kyoto Encyclopedia of Genes and Genomes) pathway database to determine the pathways associated with proteins that show differences in the supernatant of the hTERT-transduced derivatives compared to their primary parental cells. The proteins contained in the supernatant were additionally filtered for actively secreted proteins by comparison with published data from human protein databases (UniProt, Human Protein Atlas).

### 2.11. Analysis of Effects on Tumor Cell Growth Mediated by Conditioned Medium

Generation of CM and quantification of the effect on tumor cell growth was performed as explained in a recent publication [9]. Briefly, the cell cultures with a confluence of about 80% were incubated in fresh RPMI-1640 medium with 10% FBS for 72 h. For CM harvest, the supernatant was collected from the flasks, centrifuged to remove remaining cell debris, and stored at −80 °C until use. Stable green fluorescence protein (GFP) expression was introduced in the tumor cells using retroviral transduction as recently described [9]. GFP-tagged tumor cells were seeded in 48-well plates (300 tumor cells per well) and treated for 30 h with a mix of CM and fresh growth medium in a ratio of 1:1 (500 μL of each per well). Fresh growth medium alone was used as control and all experiments were performed in triplicates. Fluorescence images of the wells were automatically taken every 5 h using the IncuCyte S3 Live-cell Analysis System (Sartorius, Göttingen, Germany). Definiens Developer XD Software (Definiens, Carlsbad, CA, USA) was used to perform automated counting of the tumor cells on the images based on cell size and GFP signal. To quantify the effect of CM on the tumor cells, the cell number percentages compared to 0 h were calculated for the different time points.

### 2.12. Statistical Analysis

The data of the study were statistically analyzed using GraphPad Prism 8.0 (Graph-Pad Software, San Diego, CA, USA). Each experiment contains data of at least three independent biological replicates shown as means with SEM unless stated otherwise. The statistical differences were evaluated by Student’s *t*-test or ANOVA for comparison of two or multiple groups, respectively, and considered significant at a *p*-value < 0.05.

## 3. Results

### 3.1. Transduction with hTERT Retrovirus Does Not Change the Morphology of Mesothelial Cells or Mesothelioma-Associated Fibroblasts but Reduces Doubling Times

In a first experiment, we compared the gene expression profile of NP2 mesothelial cells with the expression profiles of the widely used mesothelial cell line Met5A [13] and of the two recently published Meso-CAF cultures Meso109F and Meso125F [9]. As expected, unsupervised clustering grouped together the two mesothelial cultures, on the one hand, and the two fibroblast cultures, on the other (Figure 1A). While Meso109F and Meso125F showed a very high degree of similarity, there were more differences between Met5A and NP2, which may reflect the fact that Met5A, in contrast to the primary NP2 cells, are SV40-transformed. Nevertheless, both Met5A and NP2 express multiple mesothelial markers and lack fibroblast marker expression (Appendix A) demonstrating their mesothelial cell origin. Meso-CAFs, in contrast, express multiple fibroblast markers but mostly lack mesothelial marker gene expression (Appendix A, Appendix A). Gene expression differences of Meso-CAFs compared to PM cells, normal lung fibroblasts, and CAFs from other tumor entities have been recently described [9] and were not explored here.

Next, we compared the morphology of NP2 that had been transduced with the hTERT-EGFP retrovirus (NP2-hT^+^) to the parental NP2 cells. After >10 passages following transduction, the overwhelming majority of NP2-hT^+^ cells showed EGFP expression, but no obvious changes in cell morphology were observed (Figure 1B). A very high percentage of EGFP-expressing cells, but no morphological changes compared to the parental counterparts, was also observed for both hTERT-transduced Meso-CAFs, Meso 109F-hT^+^, and Meso125F-hT^+^ (Figure 1C). The morphology of the commonly used mesothelial cell model Met5A and the two normal lung fibroblast models MRC-5 and Wi38 is shown in Appendix A. The identity of the hTERT-transduced derivatives with their primary cell counterparts was confirmed by short tandem repeat (STR) analysis (Table 1).

A notable difference was found with respect to doubling times. In all three hTERT-transduced cell models, a trend towards reduced doubling times was found, which was statistically significant in NP2-hT^+^ and Meso125F-hT^+^ compared to NP2 and Meso125F, respectively. In NP2, hTERT expression reduced the doubling time from <150 h to <50 h, a value similar to that found in Met5A (Table 1).

### 3.2. Transduction with hTERT Retrovirus Leads to Robust hTERT Expression, Telomerase Activity, and Lengthening of Telomeres

In order to determine the degree of hTERT overexpression after transduction with the hTERT retrovirus, we analyzed hTERT mRNA. In all cell models, hTERT transcript was strongly increased following transduction (Figure 2A). In Met5A, which was immortalized with SV40, hTERT expression was much lower than in the hTERT-transduced cell models but higher than in all primary cultures (NP2, Meso109F, Meso125F, MRC-5, Wi38). hTERT protein was also detectable in the nuclear protein fractions of all hTERT-transduced cell models in a proteome analysis (Appendix A, compare next section) but not in the nuclear proteome of the primary cells. With respect to telomerase activity, hTERT retroviral transduction led to a strong increase compared to all primary cells as determined by qPCR-TRAP assay (Figure 2B) and is additionally shown on polyacrylamide gels (Figure 2C, Appendix A). Nevertheless, the highest overall hTERT activity was found in Met5A. Mean values of the absolute telomere lengths of bulk cells were determined and showed a clear but rather moderate difference between parental primary cultures and hTERT-transduced cells (Figure 2D). Interestingly, Met5A had very short telomeres despite its high telomerase activity.

### 3.3. NP2-hT^+^ Show a High Similarity in Their Proteome and Secretome When Compared to Primary Mesothelial Cells

To compare NP2-hT^+^ with the parental NP2 in an unbiased approach, a proteome analysis was performed. Identified proteins are shown in Appendix A. Pearson correlation coefficients were 0.83, 0.87, and 0.79 for the supernatant, cytoplasmic, and nuclear fractions, respectively, between NP2-hT^+^ and NP2 (Figure 3A, Appendix A, respectively). This compared to correlation coefficients of 0.66, 0.71, and 0.56 between the supernatant, cytoplasmic, and nuclear protein fractions of NP2 versus Met5A (Figure 3B, Appendix A, respectively) and indicated a much closer relationship between parental and hTERT-transduced mesothelial cells than between mesothelial cells from different origins. Venn diagram analysis demonstrated that a high number of proteins in the supernatant (548, 41.5%) were common to NP2, NP2-hT^+^, and Met5A (Figure 3C). The smallest overlap in secreted proteins was seen between NP2 and Met5A (609), whereas NP2 and NP2-hT^+^ shared 664 proteins and NP2-hT^+^ and Met5A, both representing cell lines (as compared to the primary NP2), shared 754 secreted proteins. A volcano plot further confirmed the close link between NP2-hT^+^ and NP2 by indicating only a small number of significant differences (Figure 3D).

### 3.4. Meso109F-hT^+^ and Meso125F-hT^+^ Show a High Similarity in Their Proteome and Secretome When Compared to Primary Mesothelioma-Associated Fibroblasts

Similar results as those for NP2 and NP2-hT^+^ cells were also obtained when Meso109F-hT^+^ and Meso125-hT^+^ were compared to their respective untransduced counterparts. Identified proteins are listed in Appendix A. Correlation coefficients were 0.83, 0.92, and 0.89 for the supernatant, cytoplasmic, and nuclear fractions, respectively, for Meso109F-hT^+^ versus Meso109F, and 0.88, 0.91, and 0.87 for the respective protein fractions of Meso125F-hT^+^ versus Meso125F (Figure 4A,B, Appendix A). In the Venn diagram of proteins in the supernatant, 47.1% (1012 proteins) were shared by all Meso-CAF models, 1148 proteins by Meso109F-hT^+^ and Meso109F, and 1265 proteins by Meso125F-hT^+^ and Meso125F. A similar degree of shared proteins was also found between the two primary Meso-CAF models (1229 proteins) and between the two hTERT-transduced models (1148 proteins) (Figure 4C).

Despite the high similarity between the hTERT-transduced and untransduced cells, notable numbers of proteins were either exclusively identified or significantly altered in all models. To focus on proteins commonly affected by hTERT transduction, we analyzed changes across both hTERT^+^ versus primary Meso-CAFs or across all three hTERT^+^ cell lines. This indicated 36 protein differences (7 gained, 29 lost, 0 differentially expressed proteins) between hTERT^+^ and primary Meso-CAFs and no common changes across all three models (Appendix A). GO term analysis using these proteins revealed only a few associations with pathways of the KEGG database (Appendix A).

In an unsupervised clustering analysis performed with all biological and technical replicates of all cell models used in the study, three major branches were observed: one containing all mesothelial cells (NP2, NP2-hT^+^, Met5A), one containing all Meso-CAFs (Meso109F, Meso109F-hT^+^, Meso125F, Meso125F-hT^+^), and one containing the normal lung fibroblasts (Figure 4D).

### 3.5. Mesothelioma-Associated Fibroblasts but Not Mesothelial Cells Enhance the Growth of Human Pleural Mesothelioma Cells

The above data indicated that hTERT-transduced mesothelial cells and Meso-CAFs remain close to their parental cells representing the same cell type, respectively. To test this on a functional level, we explored the ability of CM of the primary and hTERT-transduced cell models to stimulate the growth of PM cells. Growth stimulation of cancer cells is a hallmark of CAFs [39] and has recently been demonstrated for primary Meso-CAFs [9]. Indeed, CM of both Meso109F-hT^+^ and Meso125F-hT^+^ stimulated the growth of the PM cell lines SPC212 and MSTO-211H to the same extent as CM of Meso109F and Meso125F (Figure 5A,B). In notable contrast, neither the Met5A-, NP2-, nor NP2-hT^+^-derived CM showed a significant growth stimulation of either SPC212 or MSTO-211H (Figure 5C,D). This suggests that the ability to stimulate PM growth is a specific feature of Meso-CAFs which is not shared by mesothelial cells. In addition, it demonstrates functional closeness of the hTERT-transduced cell models to their respective parental cell counterparts.

Finally, we examined the secretomes of the investigated Meso-CAFs and mesothelial cells for differences that could explain the lack of stimulation by mesothelial cell CM. Several growth factors and extracellular proteases like, for instance, the chemokine C-X-C motif chemokine ligand 12 (CXCL12), the insulin-like growth factor (IGF) 2, the WNT ligand 5B (WNT5B), and the matrix metalloproteinase 2 (MMP2) were either exclusively identified in Meso-CAF-derived CM or showed significantly lower expression in mesothelial cell-derived CM (Figure 5E).

## 4. Discussion

Numerous publications highlight the key role of the unique inflammatory TME in PM as an important driver of tumor development and progression [6,40]. Due to a lack of suitable cell models, however, many key aspects of the cellular interactions within the TME of PM have not yet been investigated in detail. A thorough characterization of mesothelioma-associated fibroblasts (Meso-CAFs) isolated from tumor tissue of PM patients has just recently been published for the first time [9]. Secreted signaling molecules of Meso-CAFs are able to highly stimulate several features of PM aggressiveness and promote the malignant transformation of pleural mesothelial cells by activating pro-oncogenic signaling pathways [9,10].

In contrast to Meso-CAFs, mesothelial cells have already been investigated by several previous studies. Mesothelial cells are generally assumed to exhibit high variability and possess extensive inherent plasticity [41,42,43]. The cells from different anatomical localizations, such as peritoneal, pleural, visceral, or parietal sites, exhibit varying morphological characteristics, gene and protein expression patterns, and functional properties [44]. Similar to mesothelioma cells, they can display different morphologies ranging from a “cobblestone”-like epithelial to a fibroblast-like appearance [43]. A recent study describes the derivation of immortalized peritoneal mesothelial cells with epithelial, fibroblast-like, and intermediate appearances from the same parental primary culture, reflecting the high plasticity of mesothelial cells [12]. Compared to Met5A, NP2 cells exhibit a rather fibroblast-like appearance. Increased passage numbers or exposure to growth factors, such as the transforming growth factor beta 1 (TGF-β 1) or the epidermal growth factor (EGF), have been identified as potential inducers of a morphologic switch towards a fibroblastic phenotype in mesothelial cells [41]. However, the phenotype of NP2 is neither a result of prolonged passaging nor dependent on the EGF contained in the ACL medium. NP2 cells already exhibited a fibroblast-like morphology directly after isolation from human tissue and no morphologic change could be observed in RMPI medium without growth factor supplementation. Despite their fibroblastoid appearance, high expression of common mesothelial cell marker genes, such as calretinin (CALB2), Wilms tumor 1 (WT1), or leucine rich repeat neuronal 4 (LRRN4), and the lack of multiple fibroblast markers clearly identify them as mesothelial cells and discriminate them from fibroblasts [44,45].

Hence, the differences in the morphology and the gene expression profiles of NP2 and Met5A cells may reflect the high variability of distinct mesothelial strains and emphasizes the need for a variety of novel pleural mesothelial cell models to enable an accurate recapitulation of the situation in humans. However, the use of the SV40 T-antigen for the immortalization of Met5A may have also contributed to these differences. In contrast to SV40-mediated immortalization, immortalization with exogenous hTERT has been postulated to retain telomerase activity in primary human cells and prevent senescence without altering phenotypic properties or causing cancer-associated changes such as the induction of cell survival pathways or the activation of cell growth-favoring receptors [46,47]. Accordingly, the authors of a recent publication describing the immortalization of pleural mesothelial cells with an epithelial phenotype via lentiviral hTERT transduction did not observe morphological changes between primary and immortalized cells [40]. Also, in our study, the transduction of hTERT in the primary NP2 cells and the primary Meso-CAFs did not elicit visible morphological changes in any of the transduced cell lines.

All our cell models showed enhanced hTERT expression and activity after retroviral transduction. Comparing protein fractions of the nucleus, the cytoplasm, and the supernatant between primary and corresponding transduced cells revealed a high degree of similarity in all of the pairs, suggesting that the retroviral transduction of hTERT did not induce widespread molecular changes in the cells on the protein level. Moreover, secreted signaling molecules derived from primary and hTERT-transduced Meso-CAFs highly stimulated PM cell proliferation to a similar extent, whereas the CM from primary and transduced NP2 cells did not stimulate tumor cell expansion. Likewise, the CM of Met5A, despite a slight trend, did not significantly increase growth of the PM cells.

Normal mesothelial cells are able to release various cytokines, chemokines, and growth factors, mainly in response to inflammatory signals or tissue injury, but have, generally, not been described to promote tumor cell growth [41,43,44]. Studies, rather, attribute inhibitory effects on adhesion and invasion of tumor cells to them, as shown in ovarian cancer [48]. Nevertheless, inflammatory signals can induce an upregulation of adhesion molecules on mesothelial cells that increase tumor cell attachment and, thus, may promote migration through the mesothelium to invade distant sites [49]. Moreover, mesothelial cells can undergo mesothelial-mesenchymal transition (MMT) induced by secreted factors from tumor cells [50]. The resulting cancer-associated mesothelial cells (CAMs) in turn secrete chemokines that can promote metastasis of cancer cells [48]. High CAM-derived WNT5A levels in ascites fluid have been shown to strongly increase the metastatic potential of ovarian cancer cells by activating its downstream effector Src family kinase Fgr [48].

To identify signaling molecules that possibly explain the disparity of growth stimulation between Meso-CAFs and mesothelial cells, we compared the secretomes of both cell types and selected potential candidates. Meso-CAFs secrete a larger number of different proteins compared to the pleural mesothelial cells. This includes several signaling molecules previously described as mediators of tumor growth stimulation that are either absent in the secretomes of the mesothelial cells or secreted to a significantly lower extent. The bone morphogenetic protein (BMP) antagonist Gremlin 1 (GREM1) can promote tumor cell proliferation and was found to be widely expressed by cancer-associated stromal cells of various human carcinomas [51]. Particularly elevated expression was found in CAFs, and study data from basal cell carcinoma even suggest that BMP antagonists secreted by CAFs are required for tumor cells to maintain their expansion [51]. IGF2 is highly upregulated in CAFs of colorectal cancer compared to normal fibroblasts and has been described as a prominent secreted factor that significantly enhances growth, migration, and invasion of tumor cells in different cancers [52,53]. Insulin-like growth factor binding protein 5 (IGFBP5) is secreted by cancer-associated stromal cells and tumor cells of various types, and its cleavage by proteases such as MMP2 induces a release of IGFs that activate proliferative and antiapoptotic signaling pathways in diverse TME components, including tumor cells or other fibroblasts [50]. In colon cancer, IGFBP5 is assumed to be upstream of WNT signaling and promotes cell growth via activation of the WNT pathway [54]. Other CAF-derived molecules, such as the chemokine CXCL12 or the extracellular protease MMP2, have also been shown to promote the proliferation and invasion of tumor cells in diverse cancers [55].

In our recent study, we have identified WNT and c-Met/Phosphoinositide 3-kinase (PI3K) signaling as critical mediators of PM cell growth stimulation by Meso-CAFs [9]. Various studies demonstrate that CAF-derived hepatocyte growth factor (HGF), the ligand of c-Met, can enhance proliferation and migration of tumor cells in different cancers [39]. Indeed, the secretomes of the pleural mesothelial cells mostly contain lower HGF levels compared to those of Meso-CAFs, but the observed differences were likely not strong enough to assume that HGF is the sole underlying cause for the strong discrepancy of CM effects between Meso-CAFs and mesothelial cells. Marked differences can be observed in the secretion of the two WNTs, WNT5A and WNT5B. WNT5B is exclusively secreted by Meso-CAFs, and the amounts of WNT5A released by mesothelial cells were also low or undetectable. CAF-derived WNT5A has recently been highlighted to play a crucial role in colorectal cancer progression by increasing tumor cell proliferation and migration [56]. In summary, we identified several secreted molecular candidates with known growth-stimulating potential either exclusively secreted by Meso-CAFs or significantly upregulated in their secretomes compared to pleural mesothelial cells. It is likely that a combination of several factors contributes to the stimulation of tumor cells by Meso-CAF-derived CM, rather than one molecule being the sole mediator of the effect.

## 5. Conclusions

Taken together, our study describes the generation and characterization of hTERT-transduced cell models and their suitability for TME research in PM. The two Meso-CAF lines and the pleural mesothelial cell model exhibit increased hTERT gene expression and protein activity resulting in longer telomeres. All transduced cell cultures preserved various features of their primary counterparts regarding morphology, protein expression, and functionality of secreted molecules. The new cell models represent valuable tools to investigate cellular interactions involved in the development, progression, and drug response of PM. Using the generated models, we demonstrate a key difference in the ability to stimulate PM growth between mesothelial cells and Meso-CAFs. A better understanding of the crosstalk between TME components in PM may help to develop new therapeutic strategies.

## Figures and Tables

**Figure 1 cells-12-02006-f001:**
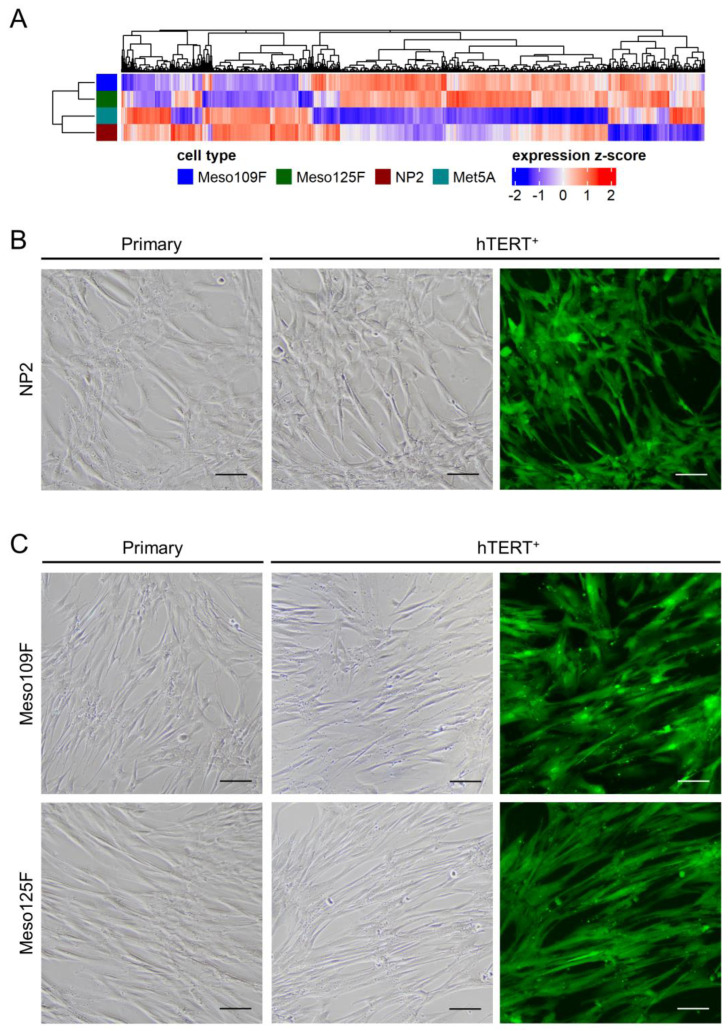
Gene expression profiles of mesothelial cells (NP2, Met5A) and Meso-CAFs (Meso109F, Meso125F) differ from each other, and transduction with hTERT-EGFP (hTERT^+^) did not induce changes in cell morphologies. (**A**) Unsupervised clustering of primary mesothelial cells and primary Meso-CAFs. (**B**) Micrographs of cultured primary NP2 cells (**left**) and the transduced NP2 derivatives (NP2-hT^+^) (**right**). (**C**) Micrographs of cultured primary Meso-CAFs (**left**) and the transduced Meso-CAFs derivatives (109F-hT^+^, 125F-hT^+^) (**right**). Scale bar = 100 µm (80×).

**Figure 2 cells-12-02006-f002:**
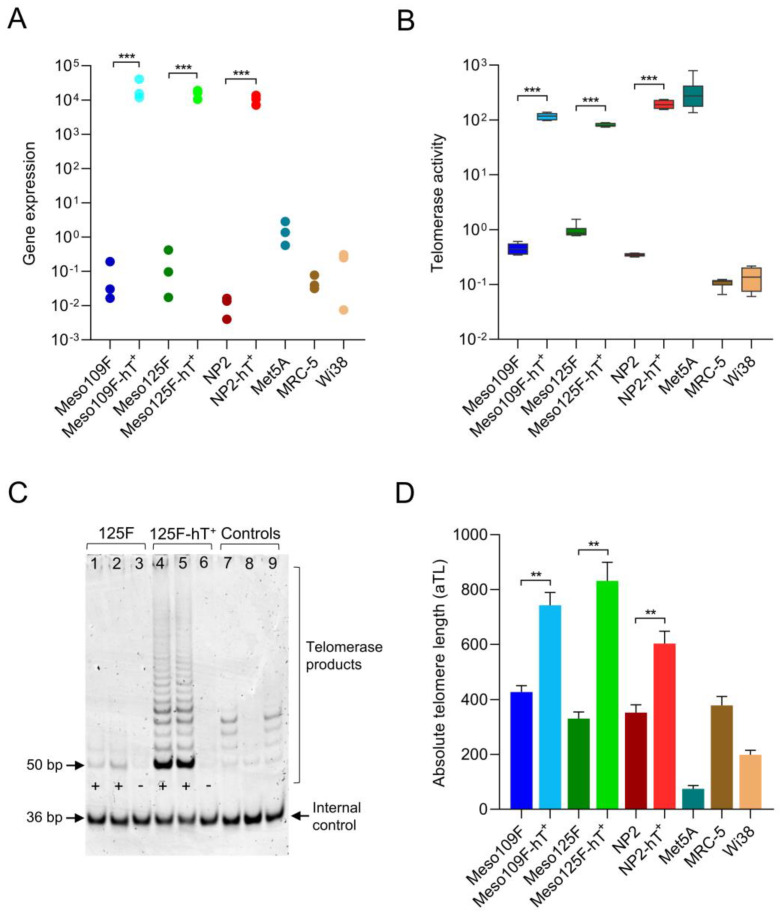
Transduction with hTERT retrovirus leads to robust hTERT expression, increased telomerase activity, and lengthening of telomeres. (**A**) Levels of hTERT gene expression in the indicated cell lines quantified by qPCR. Each dot represents one biological replicate. Gene expression data were normalized to GAPDH and β-actin expression and are shown as 2^−ΔCt^ × 10^5^. (**B**) Levels of telomerase activity in the indicated cell lines determined by qPCR-TRAP assay. Enzyme activity is shown as log_10_ of total product generated (TPG) units. (**C**) Exemplary polyacrylamide gel image showing the telomerase activity of Meso125F (125F) cells (lane 1–3) and its transduced derivative (125F-hT^+^) (lane 4–6) determined by PAGE-TRAP assay. Positive control: telomerase substrate oligonucleotide with eight telomeric repeats (TSR8) (lane 7, 9). Negative control: Mere CHAPS lysis buffer (lane 8). + probes without denaturation, − probes denaturated by heat. Probes and TSR8 positive controls were subjected to the gel in duplicates. (**D**) Absolute telomere lengths (aTLs) of the indicated cell lines measured by qPCR with oligomer standards. Telomere lengths are calculated and shown as kb per diploid genome. ** *p* < 0.01, *** *p* < 0.001. Unpaired *t*-test with Welch’s correction was used for statistical testing and all experiments were conducted in three biological replicates unless indicated otherwise.

**Figure 3 cells-12-02006-f003:**
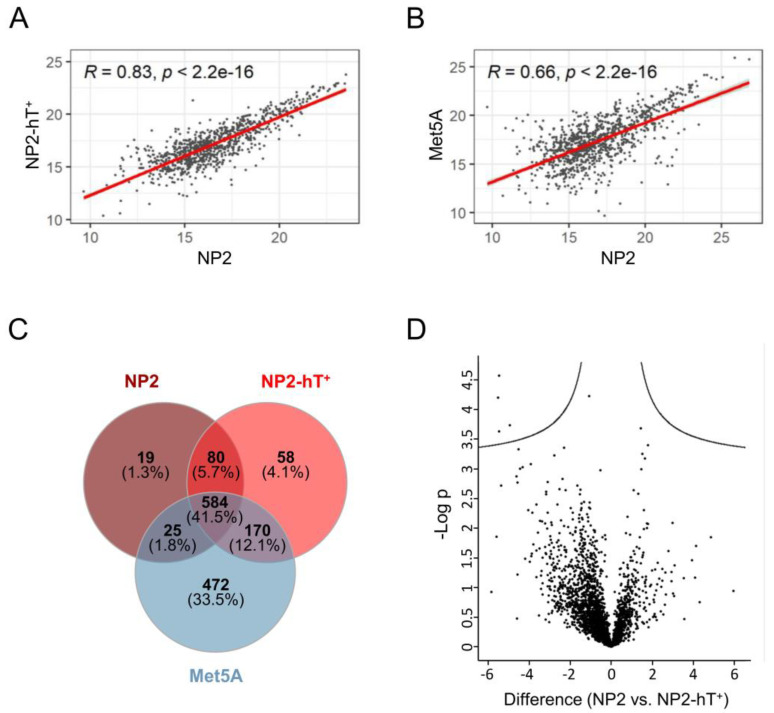
The proteins in the supernatant of the primary mesothelial cell line NP2 and its transduced derivative NP2-hT^+^ show a high similarity but clearly differ from the mesothelial cell line Met5A. (**A**,**B**) Scatter diagrams of common proteins in the supernatant between primary NP2 cells and (**A**) NP2-hT^+^ cells or (**B**) Met5A cells. Calculated Pearson correlation coefficients (R) and respective *p*-values are shown. (**C**) Venn diagram showing the numbers and overlaps of proteins identified in the supernatants of NP2, NP2-hT^+^, and Met5A. (**D**) Volcano plots of differentially expressed supernatant proteins in NP2 versus NP2-hT^+^.

**Figure 4 cells-12-02006-f004:**
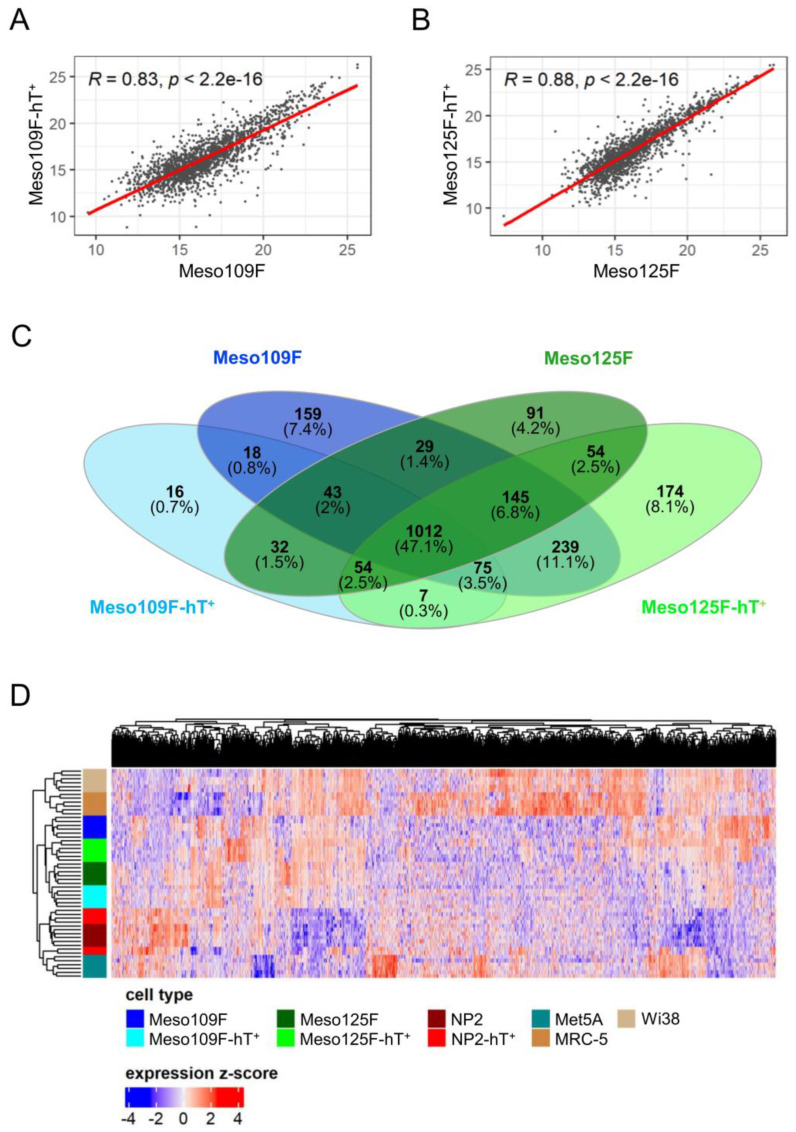
The protein composition in the supernatant of primary Meso-CAFs (Meso109F, Meso125F) and their transduced derivatives (Meso109F-hT^+^, Meso125F-hT^+^) show a high similarity. (**A**,**B**) Scatter diagrams of common proteins in the supernatant between (**A**) Meso109F and Meso109F-hT^+^ cells and (**B**) Meso125F and Meso125F-hT^+^ cells. Calculated Pearson correlation coefficients (R) and respective *p*-values are shown. (**C**) Venn diagram showing the numbers of proteins identified in the supernatants of the indicated cells lines. (**D**) Unsupervised clustering of proteins identified in the supernatant of Meso-CAFs (Meso109F, Meso125F, Meso109F-hT^+^, Meso125F-hT^+^), mesothelial cells (NP2, NP2-hT^+^, Met5A), and normal lung fibroblasts (MRC-5, Wi38) is depicted in a heatmap.

**Figure 5 cells-12-02006-f005:**
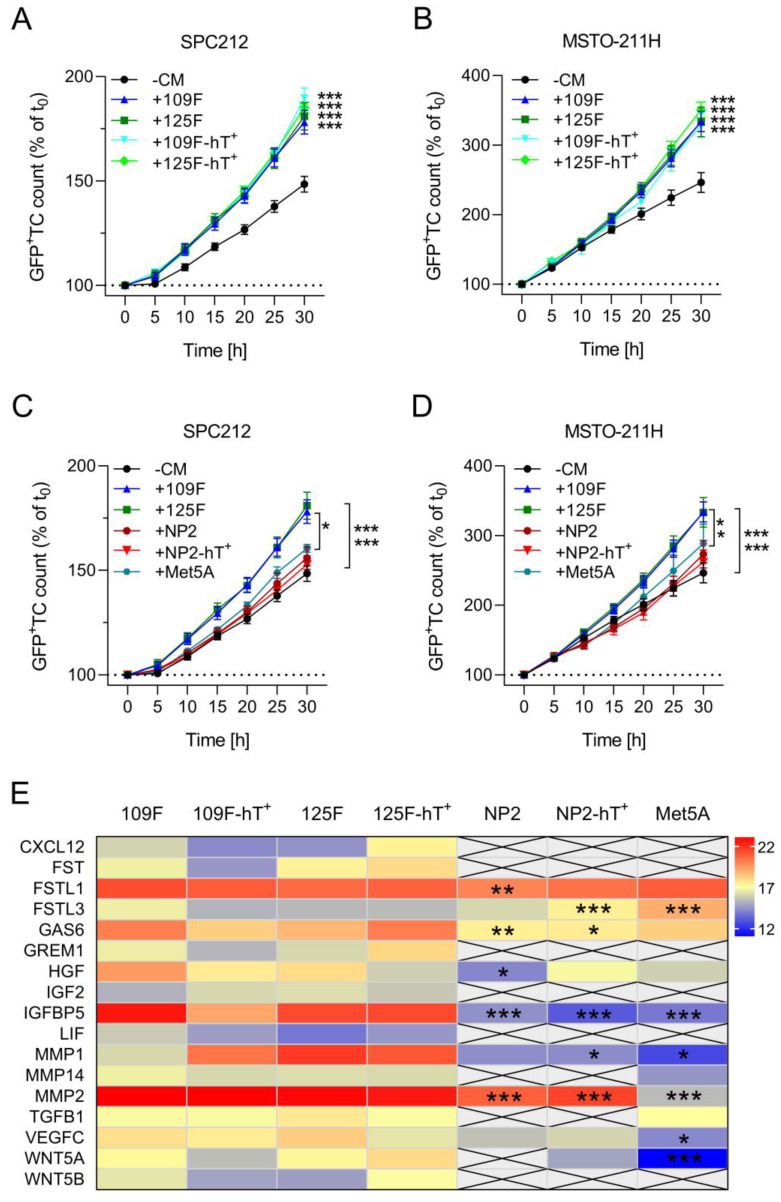
Conditioned medium of primary and hTERT-transduced Meso-CAFs but not mesothelial cells enhance the growth of human pleural mesothelioma cells. (**A**–**D**) Green fluorescent protein-tagged (GFP^+^) SPC212 (**A**,**C**), and MSTO-211H (**B**,**D**) cells were incubated without conditioned medium (−CM) or with medium conditioned by Meso109F (+109F), Meso125F (+125F), Meso109F-hT^+^ (+109F-hT^+^), Meso125F-hT^+^ (+125F-hT^+^), NP2 (+NP2), NP2-hT^+^ (+NP2-hT^+^), or Met5A (+Met5A). Micrographs were taken every 5 h and numbers of GFP^+^ tumor cells (TC) were determined by automated image analysis. * *p* < 0.05, *** *p* < 0.001. TC number in the presence versus absence of CM or as indicated by brackets; two-way ANOVA with Tukey’s multiple comparisons test. (**E**) Heatmap showing levels of detected proteins (LFQ values) secreted by the indicated cells. * *p* < 0.05, ** *p* < 0.01, *** *p* < 0.001. Protein level versus the averaged Meso-CAF protein level; two-way ANOVA with Dunnett’s multiple comparisons test.

**Table 1 cells-12-02006-t001:** Short tandem repeat (STR) profiles and mean doubling times (in hours) with standard deviations (SDs) of the indicated cells. * *p* < 0.05, *** *p* < 0.001. hTERT-transduced (hT^+^) cells versus the corresponding parental primary counterpart. Unpaired *t*-test was used for statistical testing.

	Meso 109F	Meso 109F-hT^+^	Meso 125F	Meso 125F-hT^+^	NP2	NP2-hT^+^	Met5A	MRC-5	Wi38
**Repeat markers**									
D3S1358	15	15	17	17	15	15	17/18/19	15/17	16/17
TH01	8/9	8/9	7/9	7/9	6/9.3	6/9.3	6/9.3	8	8/9.3
D21S11	28/29	28/29	31.2/32.2	31.2/32.2	28/31	28/31	31.2	31.2	30/30.2
D18S51	12/19	12/19	14/15	14/15	14/16	14/16	15/23	15/21	16/18
Penta_E	13/17	13/17	5/8	5/8	11/16	11/16	10/16	12/16	13/14
D5S818	12/13	12/13	12	12	12/13	12/13	12	11/12	10
D13S317	11/13	11/13	10/12	10/12	8/11	8/11	11/13	11/14	11
D7S820	9/12	9/12	7/8	7/8	9/11	9/11	10	10/11	9/11
D16S539	13	13	11/12	11/12	11	11	12	9/11	11/12
CSF1PO	10/12	10/12	12	12	12/14	12/14	10/12	11/12	10/12
Penta_D	9	9	13	13	9/12	9/12	10/11	12	13
AMEL	X	X	X/Y	X/Y	X/Y	X/Y	X/Y	X/Y	X
vWA	17/19	17/19	16	16	14/16	14/16	15/18	15	19/20
D8S1179	12/15	12/15	12/15	12/15	12/13	12/13	12/13	13	14
TPOX	8/11	8/11	8/10	8/10	8	8	8	8	8
FGA	19/23	19/23	21/23	21/23	22/24	22/24	22/23	21/23	22/24
**Doubling times (in hours)**									
Mean	161.5	114.4	180.5	109.1 *	162.1	43.4 ***	40.9	227.9	181.3
SD	41.35	7.03	37.40	5.44	14.71	14.11	11.07	29.05	35.23

## Data Availability

All datasets generated and analyzed during the current study are available in the following repositories under the respective identifiers. Proteomics Identifications Database (PRIDE) (https://www.ebi.ac.uk/pride (accessed on 4 July 2023)): PXD035987, PXD036017, PXD036127, PXD043511, and PXD043534; ArrayExpress (https://www.ebi.ac.uk/biostudies/arrayexpress (accessed on 4 July 2023)): E-MTAB-12177, E-MTAB-8986. An overview of mass spectrometry datasets, including sample names, corresponding file names, and PRIDE accession numbers are shown in Appendix A.

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
