# Peer review of "Primary and hTERT-Transduced Mesothelioma-Associated Fibroblasts but Not Primary or hTERT-Transduced Mesothelial Cells Stimulate Growth of Human Mesothelioma Cells"

_cells, 2023, doi:10.3390/cells12152006_

Round 1

Reviewer 1 Report

Title: Primary and hTERT-Transduced Mesothelioma-Associated Fibroblasts but not Primary or hTERT-Transduced Mesothelial Cells Stimulate Growth of Human Mesothelioma Cells

ID: cells-2524949

Article Type: Article

Journal: Cells

Section: Cellular Pathology

The aim of the present article was to describe hTERT-transduced Meso-CAFs and normal pleural mesothelial cells, and investigate their ability to stimulate PM cell growth.

The work is very interesting and exhaustive. It is written very well, the concepts are clearly explained. For me there are no relevant changes to make.

Introduction

Please the authors integrate with the information that other fibers are risk factors for the onset of pleural mesothelioma. The following research work may help: Filetti V, Loreto C, Falzone L, Lombardo C, Cannizzaro E, Castorina S, Ledda C, Rapisarda V. Diagnostic and Prognostic Value of Three microRNAs in Environmental Asbestiform Fibers-Associated Malignant Mesothelioma. J Pers Med. 2021 Nov 15;11(11):1205. doi: 10.3390/jpm11111205. PMID: 34834557; PMCID: PMC8618926.

Materials and Methods

Please the authors write the company of ACL medium and RPMI-1640, DMEM and MEM medium.

Please the authors write the specific protocols used after the used kits, in particular the versions of handbooks.

Figure 1

Please the authors add the magnification values of micrographs.

Author Response

Reviewer 1: General comments to the authors:

The aim of the present article was to describe hTERT-transduced Meso-CAFs and normal pleural mesothelial cells, and investigate their ability to stimulate PM cell growth. The work is very interesting and exhaustive. It is written very well, the concepts are clearly explained. For me there are no relevant changes to make.

Author response: We thank the reviewer for the positive comments on our manuscript and the constructive suggestions, which we have addressed below.

Major suggestions:

Reviewer 1: There are no major issues with the manuscript.

Minor suggestions:

Reviewer 1: Introduction: Please the authors integrate with the information that other fibers are risk factors for the onset of pleural mesothelioma. The following research work may help: Filetti V, Loreto C, Falzone L, Lombardo C, Cannizzaro E, Castorina S, Ledda C, Rapisarda V. Diagnostic and Prognostic Value of Three microRNAs in Environmental Asbestiform Fibers-Associated Malignant Mesothelioma. J Pers Med. 2021 Nov 15;11(11):1205. doi: 10.3390/jpm11111205. PMID: 34834557; PMCID: PMC8618926.

Author response: We thank the reviewer for the suggestion. We have included the information that fibers other than asbestos also represent a considerable risk factor for pleural mesothelioma development into the introduction of the manuscript, and cite the suggested reference. 

Reviewer 1:  Materials and Methods: Please the authors write the company of ACL medium and RPMI-1640, DMEM and MEM medium. Please the authors write the specific protocols used after the used kits, in particular the versions of handbooks.

Author response: We have added the requested information on the used growth media and the methodological protocols in the materials and methods part. The ACL medium was mixed from the ingredients shown in Supplementary Table S1 and this is now mentioned in the text.

Reviewer 1:  Figure 1: Please the authors add the magnification values of micrographs.

Author response: All micrographs were taken at the same magnification, and in the original version of the manuscript a scale bar was shown in one representative micrograph of each figure panel. For increased clarity, the scale bar has now been added to each individual micrograph and the magnification value has been added to the figure legend (Figure 1B, Figure 1C). This was also done for the Supplementary Figure S2.

Reviewer 2 Report

General:

Ries et al. present a well-written paper on the immortalization of mesothelioma-associated fibroblasts and primary mesothelial cells by retroviral hTERT-transduction and study their influence on the growth of mesothelioma cells. 

The topic is important for understanding the biology of mesothelioma and to develop new treatments for this still incurable disease. The immortalization of mesothelial cells with hTERT is also a welcome addition to the tissue culture “toolbox”, reducing the dependency on primary cultures or SV40-transformed cell lines.

Major:

There are no major issues with the manuscript.

Minor:

Figure 1A shows the expression profiles of four cell lines but without individual gene names (because of the size limitation), while Supplementary Figure S1 provides only a selection of differentially expressed genes. It would be nice if the authors could provide a supplement either with a detailed version of Figure 1A or a table with all gene names and their relative expression.

In Figure 5C & D (growth stimulation by conditioned medium of different cell lines) there appears to be some stimulation by Met5A, despite being significantly less then by the two Meso-CAFs. Was the Met5A effect, compared to “-CM”, statistically significant? If yes, please include this in your discussion regarding Met5A as a non-malignant mesothelial cell line because the effects of SV40 transformation most likely go beyond mere immortalization (the proteomic profile of Met5A in comparison to NP2 / NP2-hT+ already points in that direction).

Author Response

Reviewer 2: General comments to the authors:

Ries et al. present a well-written paper on the immortalization of mesothelioma-associated fibroblasts and primary mesothelial cells by retroviral hTERT-transduction and study their influence on the growth of mesothelioma cells. The topic is important for understanding the biology of mesothelioma and to develop new treatments for this still incurable disease. The immortalization of mesothelial cells with hTERT is also a welcome addition to the tissue culture “toolbox”, reducing the dependency on primary cultures or SV40-transformed cell lines.

Author response: We thank the reviewer for the positive comments on our manuscript and the suggestions, which we have addressed below.

Major suggestions:

Reviewer 2: There are no major issues with the manuscript.

Minor suggestions:

Reviewer 2: Figure 1A shows the expression profiles of four cell lines but without individual gene names (because of the size limitation), while Supplementary Figure S1 provides only a selection of differentially expressed genes. It would be nice if the authors could provide a supplement either with a detailed version of Figure 1A or a table with all gene names and their relative expression.

Author response: We thank the reviewer for the suggestion. As suggested, we now include an additional Supplementary Table (Supplementary Table S2) showing the data of all oligos of annotated genes from the gene expression microarrays, which were used to generate the heatmap in Figure 1A. Supplementary Table S2 includes the gene names, a short gene description and the respective raw and normalized expression levels for each cell line shown in Figure 1A (Meso109F, Meso125F, Met5A, NP2).

Reviewer 2: In Figure 5C & D (growth stimulation by conditioned medium of different cell lines) there appears to be some stimulation by Met5A, despite being significantly less then by the two Meso-CAFs. Was the Met5A effect, compared to “-CM”, statistically significant? If yes, please include this in your discussion regarding Met5A as a non-malignant mesothelial cell line because the effects of SV40 transformation most likely go beyond mere immortalization (the proteomic profile of Met5A in comparison to NP2 / NP2-hT+ already points in that direction).

Author response: The effect on PM cell growth induced by CM of Met5A was not significantly different from -CM in both PM cell lines. We agree with the reviewer that SV40-transformation could go beyond mere immortalization, but the respective p-values calculated from our experimental data (n=10) only indicate a slight trend and no significant difference. The trend is now mentioned in the discussion section, but due to the absence of a significant effect, we think it is appropriate to not speculate about possible causes.

Reviewer 3 Report

Evidence has shown that cancer-associated fibroblasts (CAF) play critical roles in maintaining the immunosuppressive tumor microenvironment and promote pro-tumorigenic features including angiogenesis, tumor cell proliferation, invasion and metastasis. CAFs secrete growth factors and extracellular matrix proteins that promote the proliferation and survival of mesothelioma cells.

This study performed hTERT transduction into mesothelioma-associated fibroblasts (Meso-CAFs) to compare with primary CAF, as well as mesothelial cells with/without hTERT transduction. They found that transduction with hTERT retrovirus leads to robust hTERT expression, increased telomerase activity, and lengthening of telomeres. The conditioned medium (CM) of primary and hTERT-transduced Meso-CAFs stimulated PM cell growth to the same extent, but not mesothelial cells enhances the growth of human pleural mesothelioma cells.

This is a fascinating study. The data that authors presented in this study are robust and support their conclusion. Overall, the manuscript is well-prepared and could be publishable if a revision could be made.

Minor comments:

The authors showed strongly supporting their conclusion, however, there seems to be a lack of comparison between Meso-CAFs and primary CAFs. What is the difference in impact of hTERT transduction into Meso-CAF on PM cell growth?

Fig. 4, The protein composition in the supernatant of primary Meso-CAFs (Meso109F, Meso125F) 397 and their transduced derivatives (Meso109F-hT+, Meso125F-hT+) show a high similarity. However, in Fig 4C the difference of proteins was observed (32+43+29+91) in Meso125F and (7+75+239+174) in Meso125F-hT+, respectively. It seems that Meso125F-hT+ resulted in more difference than Meso125F in CM.

Fig. 5, It is interesting to compare the impact of each CM on mesothelioma cell growth It would be more interesting to do co-culture of Meso-CAF hT- or hT+ with mesothelioma cells to see if there is any impact on mesothelioma cell growth.

This study supports a notion that targeting mesothelioma-associated fibroblasts can be a potential therapeutic strategy for mesothelioma.

Among the difference between Meso-CAF hT- and hT+ in the heat map of Fig 4D, the up or down-regulated proteins may be involved in different protein-protein interactions (PPI). It would be good if you could look further into your data.  

The authors mainly focused on protein changes after transduction of Meso-CAFs. There might be some changes in gene expression as well, therefore, it would more interesting to look at gene signatures in future study.

Author Response

Reviewer 3: General comments:

Evidence has shown that cancer-associated fibroblasts (CAF) play critical roles in maintaining the immunosuppressive tumor microenvironment and promote pro-tumorigenic features including angiogenesis, tumor cell proliferation, invasion and metastasis. CAFs secrete growth factors and extracellular matrix proteins that promote the proliferation and survival of mesothelioma cells. This study performed hTERT transduction into mesothelioma-associated fibroblasts (Meso-CAFs) to compare with primary CAFs, as well as mesothelial cells with/without hTERT transduction. They found that transduction with hTERT retrovirus leads to robust hTERT expression, increased telomerase activity, and lengthening of telomeres. The conditioned medium (CM) of primary and hTERT-transduced Meso-CAFs stimulated PM cell growth to the same extent, but not mesothelial cells enhance the growth of human pleural mesothelioma cells. This is a fascinating study. The data that authors presented in this study are robust and support their conclusion. Overall, the manuscript is well-prepared and could be publishable if a revision could be made.

Author response: We thank the reviewer for the positive comments regarding our study and the constructive suggestions, which we have addressed below.

Major suggestions:

Reviewer 3: There are no major issues with the manuscript.

Minor suggestions:

Reviewer 3: The authors showed strongly supporting their conclusion, however, there seems to be a lack of comparison between Meso-CAFs and primary CAFs. What is the difference in impact of hTERT transduction into Meso-CAF on PM cell growth?

Author response: We thank the reviewer for the comment. As outlined below, we have increased the part of the manuscript comparing hTERT-transduced and primary CAFs. Regarding the impact of hTERT transduction into Meso-CAF on PM cell growth, we show in Figure 5A and 5B that the CM of the primary Meso-CAFs and their hTERT-transduced counterparts (Meso109F vs. Meso109F-hT+, Meso125F vs. Meso125F-hT+) stimulated PM cell growth to the same extent compared to -CM for both PM cell lines (SPC212, MSTO-211H). Thus, in the functional experiments performed, we did not identify a significant difference between the primary cells and the hTERT-transduced derivatives.

Reviewer 3: Fig. 4: The protein composition in the supernatant of primary Meso-CAFs (Meso109F, Meso125F) and their transduced derivatives (Meso109F-hT+, Meso125F-hT+) show a high similarity. However, in Fig 4C the difference of proteins was observed (32+43+29+91) in Meso125F and (7+75+239+174) in Meso125F-hT+, respectively. It seems that Meso125F-hT+ resulted in more difference than Meso125F in CM.

Author response: We agree with the reviewer that, despite of a high similarity between the primary and transduced cells, the transduction with hTERT also resulted in some changes of the protein composition compared to the parental cells. In the revised version of our manuscript, we now include a paragraph in the results section addressing the protein differences in the supernatants after hTERT-transduction. We have included an additional Supplementary Table (Supplementary Table S4) showing all proteins that were either gained, lost, or differentially expressed after hTERT-transduction across both transduced Meso-CAFs.

Reviewer 3: Fig. 5: It is interesting to compare the impact of each CM on mesothelioma cell growth. It would be more interesting to do co-culture of Meso-CAF hT- or hT+ with mesothelioma cells to see if there is any impact on mesothelioma cell growth.

Author response: In a recent publication (J Exp Clin Cancer Res. 2023; 42: 27., doi: 10.1186/s13046-022-02582-0) we have focused on characterizing the primary Meso-CAFs including their effect on several aspects of PM aggressiveness (growth, invasion, therapy response). To address these issues we used 2D-co-culture, 3D co-culture and CM-based experimental approaches. We could show that PM cell growth stimulation is to a large degree mediated by the secretome of Meso-CAFs and that the growth stimulatory effects observed in co-cultures with Meso-CAFs being present could be well reproduced using only their CM. In the current manuscript, our focus was to show that hTERT-transduction of both Meso-CAFs and mesothelial cells is a) feasible and b) retains key characteristics of the cell types - using a more limited set of experiments for functional tests. We have used CM experiments because we think, with respect to our previous data, that they represent a good method to assess the ability of other cell types to interfere with PM growth. The co-culture experiments as performed in our previous study require a clear discrimination of both cell types in the co-cultures, which in our previous study was based on stable GFP expression by PM cells and Meso-CAFs without fluorescence. Since all hTERT-transduced cells also exhibit stable GFP expression, co-culture experiments with hTERT-transduced Meso-CAFs would first require inducing the stable expression of another fluorescent marker, such as mCherry in PM cell lines. While we agree with the reviewer that co-culture experiments in different formats could potentially reveal additional information, we would kindly ask for the reviewer´s understanding that we would consider them beyond the scope of the current manuscript.

Reviewer 3: This study supports a notion that targeting mesothelioma-associated fibroblasts can be a potential therapeutic strategy for mesothelioma. Among the difference between Meso-CAF hT- and hT+ in the heatmap of Fig 4D, the up or down-regulated proteins may be involved in different protein-protein interactions (PPI). It would be good if you could look further into your data. The authors mainly focused on protein changes after transduction of Meso-CAFs. There might be some changes in gene expression as well, therefore, it would more interesting to look at gene signatures in future study.

Author response: We thank the reviewer for the suggestion and have re-analyzed our data to identify protein differences in the supernatants between hTERT+ and primary (hTERT-) Meso-CAFS. The resulting protein lists are shown in a new Supplementary Table (Supplementary Table S4). To explore possible interactions of these proteins, a GO term analysis was performed and the results are shown in Supplementary Figure S7. It has to be noted, however, that this analysis was hampered by the relatively small number of changed proteins that both Meso-CAF models had in common. No supernatant proteins were found universally affected by hTERT transduction across all three cell models (Meso109F, Meso125F, NP2) in our approach. While we think that secreted proteins are very important for tumor microenvironment research, we fully agree with the reviewer that looking into gene signatures in future studies would be very interesting and could reveal gene expression changes associated with, for instance, telomerase functions beyond immortalization.